# Autophagy-Related Gene WD Repeat Domain 45B Promotes Tumor Proliferation and Migration of Hepatocellular Carcinoma through the Akt/mTOR Signaling Pathway

**DOI:** 10.3390/diagnostics13050906

**Published:** 2023-02-27

**Authors:** Jiahao Li, Lansi Chen, Jingjing Pang, Chunxiu Yang, Wen Xie, Guoyan Shen, Hongshan Chen, Xiaoyi Li, Shu-Yuan Xiao, Yueying Li

**Affiliations:** 1Department of Pathology, Wuhan University Zhongnan Hospital, Wuhan 430071, China; 2Center for Pathology and Molecular Diagnostics, Wuhan University, Wuhan 430071, China; 3Department of Pathology, Union Hospital, Tongji Medical College, Huazhong University of Science and Technology, Wuhan 430022, China; 4Department of Pathology, Jiahui International Hospital, Shanghai 200000, China

**Keywords:** hepatocellular carcinoma, autophagy, prognosis, WDR45B, Akt/mTOR signaling pathway

## Abstract

Hepatocellular carcinoma (HCC) is a highly aggressive malignant tumor. It has been found that autophagy plays a role both as a tumor promoter and inhibitor in HCC carcinogenesis. However, the mechanism behind is still unveiled. This study aims to explore the functions and mechanism of the key autophagy-related proteins, to shed light on novel clinical diagnoses and treatment targets of HCC. Bioinformation analyses were performed by using data from public databases including TCGA, ICGC, and UCSC Xena. The upregulated autophagy-related gene WDR45B was identified and validated in human liver cell line LO2, human HCC cell line HepG2 and Huh-7. Immunohistochemical assay (IHC) was also performed on formalin-fixed paraffin-embedded (FFPE) tissues of 56 HCC patients from our pathology archives. By using qRT-PCR and Western blots we found that high expression of WDR45B influenced the Akt/mTOR signaling pathway. Autophagy marker LC3- II/LC3-I was downregulated, and p62/SQSTM1 was upregulated after knockdown of WDR45B. The effects of WDR45B knockdown on autophagy and Akt/mTOR signaling pathways can be reversed by the autophagy inducer rapamycin. Moreover, proliferation and migration of HCC can be inhibited after the knockdown of WDR45B through the CCK8 assay, wound-healing assay and Transwell cell migration and invasion assay. Therefore, WDR45B may become a novel biomarker for HCC prognosis assessment and potential target for molecular therapy.

## 1. Introduction

Liver cancer has been one of the most common and malignant tumors in the digestive system. Over the world, liver cancer ranked the 6th in the morbidity and the 3rd in the mortality of cancers, and the 5-year survival rate was only about 18% [1,2,3]. Hepatocellular carcinoma (HCC) accounts for more than 90% of primary liver cancer. It is mainly developed from chronic fibrous liver disease caused by hepatitis B virus (HBV) or hepatitis C virus (HCV) infection, alcohol consumption, and metabolic syndrome. In recent years, with the rising incidence of obesity, diabetes, and non-alcoholic fatty liver disease/non-alcoholic steatohepatitis, chronic hepatitis has been thought to be the major driver of HCC increased morbidity globally [4].

At present, conventional treatments for HCC principally involved surgical resection, radiofrequency ablation (RFA), percutaneous ethanol injection, trans-arterial chemoembolization, liver transplantation, chemotherapy, molecular targeted therapy, and immunotherapy. Unfortunately, HCC was usually diagnosed at an advanced stage and was prone to metastasis and recurrence, so the effectiveness of the conventional treatments was fairly limited. Even if receiving thorough surgical excision or RFA, the carcinogenic tissue microenvironment in the remnant liver can still cause the recurrence in 70% of patients within 5 years [5]. Sorafenib, the only first-line multi-target tyrosine kinase inhibitor for advanced HCC, can only increase the median overall survival of most advanced patients to around 1 year [6]. Moreover, drug resistance was also one of the reasons for poor prognosis in HCC patients, which always occurred in patients receiving anti-cancer drug treatments for over 6 months [7]. Therefore, it is of great significance to elucidate the underlying pathophysiological process of occurrence and development and explore the complex mechanism behind HCC metastasis, which might be helpful for the early diagnosis of HCC and identifying effective prognostic biomarkers.

Autophagy, a highly conserved catabolic process, plays a critical part in the quality control of organelles or cellular proteins, adjustment of nutrient balance, and pathogen defense. Based on the difference in morphology and mechanism, autophagy can be categorized into 3 subtypes, namely macro-autophagy, micro-autophagy, and chaperone-mediated autophagy [8,9]. The “autophagy” is usually referred to as “macro-autophagy”. In the process of autophagy, cytoplasmic macromolecules, aggregative proteins, damaged organelles, and pathogens are delivered to lysosomes through autophagosomes, which are digested by lysosomal hydrolases to generate nucleotides, amino acids, fatty acids, glucose, and ATP, which are eventually recycled into the cytoplasmic matrix. Nucleation of autophagic separation membrane, formation of autophagosome, expansion, and extension of autophagosome membrane, docking and fusion with lysosome membrane, as well as degradation and recycling of products in vesicles are the key steps in autophagy, in which there are lots of autophagy-related genes (ATGs) involved [10]. ULK1 complex consisting of unc-51-like autophagy activating kinase 1 (ULK1), ATG13, AG101, FAK family interacting protein of 200 kDa (FIP200), can all induce the nucleation of autophagic separation membrane. VPS34, beclin1, and ATG14L assembled by AMBRA1 promote the formation of autophagosomes. Later, the microtubule-associated protein light chain 3 (LC3) is recruited to help the autophagosome elongation. Lastly, under the regulation of Rabs, SNAREs and hydrolysis enzymes located in lysosomes could degrade the cargo on arrival [11,12]. Autophagy was considered controlled by a series of signal transduction, involving Serine/Threonine protein kinase (Akt)/mammalian target of rapamycin (mTOR) pathway, Liver kinase B1 (LKB1)/AMP-activated protein kinase (AMPK) pathway and Beclin complex [13].

Currently, it is widely recognized that autophagy appears to be a “double-edged sword” in HCC. On the one hand, autophagy can play a tumor suppressor role in the early stage of tumor, remove oncogenic protein aggregates and damaged organelles, alleviate oxidative stress, reduce DNA damage and genomic instability, as well as prevent the conversion of histologically normal cells into early premalignant lesions. On the other hand, under the stimulation of starvation, hypoxia, growth factor depletion, and anti-cancer treatment, autophagy can be strongly activated by removing toxic oxygen-free radicals, relieving stress, and maintaining mitochondrial function to enhance proliferation and survival of tumor cells [14]. It has been confirmed that autophagy is highly correlated with the tumor progression, tumor immune microenvironment, and drug resistance of HCC [15]. Inhibiting mTOR signaling pathway-inducing autophagy resulted in an obvious antitumor effect along with improved overall survival rates in HCC patients. The suppression of the autophagy could also benefit the HCC treatment by enhancing the response to anti-cancer agents. The autophagy inhibitor hydroxychloroquine (HCQ) combined with sorafenib has improved efficacy in HCC treatment when compared with sorafenib alone [16]. Moreover, ATGs have been defined as predictive signatures for anti-PD-L1 immunotherapy [17]. Targeting the NF-κB signaling pathway of tumor-associated macrophages, affecting the level of p62 to activate the autophagy, enabled the tumor-center T cells to restore their sensitivity to anti-PD-L1 therapy [18,19]. Therefore, a more in-depth investigation of complex and potential mechanism of autophagy in HCC would be essential for the treatment. Given the above situation, this study aimed to search for a potential biomarker and therapeutic target for HCC that is highly related to autophagy, in the hope of providing informative clues for effective management of HCC and related research.

## 2. Materials and Methods

### 2.1. Bioinformation Analysis

RNA-Seq transcription data, survival data, and clinical information were obtained from the HCC patients in TCGA (The Cancer Genome Atlas, https://portal.gdc.cancer.gov/, accessed on 30 April 2021), UCSC Xena (http://xena.ucsc.edu/, accessed on 30 April 2021) and ICGC (International Cancer Genome Consortium, https://dcc.icgc.org/, accessed on 30 April 2021). A total of 286 autophagy-related genes were collected from the HADb (Human Autophagy Database, http://www.autophagy.lu/, accessed on 4 May 2021). The raw microarray data were preprocessed using background correction and robust multi-array analysis (RMA) normalization with “Affy” R package in R version 4.2.0. software. “Limma” R package was used to screen out the differentially expressed autophagy-related genes by a cut-off of |LogFC| > 1 (fold change), *p* < 0.05. Then, univariate Cox regression analysis was performed based on the RNA-Seq transcription data and matched survival data to obtain differentially expressed autophagy-related genes highly correlated with survival. Insertion of the differentially expressed and prognostic autophagy-related genes in TCGA and ICGC datasets were taken. The protein–protein interaction network between these genes was visualized by “igraph” R package based on data obtained via “STRINGdb” R package. Filtering out those genes that have already been studied universally in HCC by retrieval in PubMed (https://pubmed.ncbi.nlm.nih.gov/, accessed on 15 June 2021), WDR45B was determined to be the target gene in this study. Later, the Kaplan–Meier survival curve and clinical traits related to the expression of WDR45B were plotted by “ggplot2” R package. Moreover, genomics characteristics of WDR45B involved somatic mutation, copy number variation, methylation level, and co-expression genes were analyzed based on the TCGA-HCC cohorts from cBioPortal for Cancer Genomics (http://www.cbioportal.org/, accessed on 2 May 2022). MethSurv (https://biit.cs.ut.ee/methsurv/, accessed on 21 August 2021) was used to compare the survival differences between high and low methylation levels. GSEA (Gene set enrichment analysis) was conducted with GSEA 4.1.0 software. WDR45B enriched KEGG pathways were picked out according to the |NES| > 1 (normalized enrichment score) and NOM (norminal) *p*-value < 0.05.

### 2.2. Immunohistochemistry (IHC)

The HCC tissue microarray was made by formalin-fixed paraffin-embedded (FFPE) specimens originating from 56 patients with primary HCC date from January 2021 to December 2021. The FFPE specimens of 3 patients with hepatic contusion were used as normal control. All the paraffin-embedded specimens were provided by the Pathology Department of Zhongnan Hospital of Wuhan University. The paraffin sections were dried at 60 °C for 1 h and dewaxed in xylene and ethanol. Citrate buffer was used to repair antigen, and then the sections were blocked by 3% hydrogen peroxide for 10 min and 5% goat serum (ZSGB-Bio, Beijing, China) for 1 h. Afterward, sections were incubated at 4 °C overnight with the WDR45B primary antibody (1:200, Signalway Antibody). On the following day, the sections were incubated with HRP-conjugated Goat Anti-Rabbit IgG H&L secondary antibody (abcam, Cambridge, MA, USA), and the DAB (Dako Diagnostics AG, Baar, Switzerland) was stained in appropriate time for microscopic examination. Washing with PBS three times was included between all processes. For imaging, the sections were scanned by an Olympus BX51 microscope equipped with a DP74 digital camera. Both the staining intensity and degree were assessed in semi-quantitative analysis by ImagePro Plus 6.0.

### 2.3. Cell Culture and Lentivirus Infection

HCC cell lines HepG2 and Huh-7 cells were purchased from Shanghai cell bank of the Chinese Academy of Science. The normal human hepatocytes LO2 cells were provided by Wuhan University Medical Science Research Center. These three types of adherent cells were cultured in DMEM supplemented with 10% FBS and 1% penicillin/streptomycin at 37 °C in a humidified atmosphere of 5% CO_2_. Cells were grown to 75% to 80% confluency and collected by trypsinization with 0.25% trypsin-EDTA.

For lentivirus infection, the HitransG (Genechem, Shanghai, China) was used according to the manufacturer’s instructions. HepG2 cells were infected at a high multiplicity of infection (MOI = 20) with LV-WDR45B-RNAi and blank lentivirus. When lentiviral infection efficacy reached around 80%, 2.0 μg/mL puromycin was used to screen the stable cells. A qRT-PCR was performed to verify the efficiency of the WDR45B knockdown.

### 2.4. Quantitative Real-Time PCR

The RNA of cells was extracted with RNA extraction kit (TIANGEN), then RNA reverse transcription PCR was carried out using reverse transcription kits (Toyobo) to obtain cDNA. The cDNA, primers, and SYBER Green Master Mix (Toyobo) were mixed for a final volume of 12 μL to perform quantitative RT-PCR analysis with the CFX connect Real-Time PCR Detection System (Bio-Rad). The primers used were as follows: WDR45B Forward primer, 5′-CGAGAAAGGGACGCTTATAAGA-3′; Reverse primer, 5′-TGATGCAGTAAATATTGGCTGC-3′; ACTB Forward primer, 5′-TCGAGTCGCGTCCACC-3′; Reverse primer, 5′-GGGAGCATCGTCGCCC-3′. Relative expression levels were calculated using 2-ΔΔCt method.

### 2.5. Western Blot

Firstly, total proteins of cells were extracted using RIPA lysis buffer and then subjected to centrifugation at 12,000× *g* rpm and 4 °C for 30 min. Subsequently, cell proteins were quantitated by BCA Protein Assay Kit (Biosharp, Tallinn, Estonia) following the manufacturer’s guidance. The protein samples were separated by using 7.5%, 10%, and 12.5% SDS polyacrylamide separating gel (EpiZyme, Cambridge, MA, USA) and electro-blotted onto PVDF membranes (Millipore, Burlington, MA, USA). After that, the membranes were blocked with 5% non-fat skimmed milk in TBST buffer for 1 h, followed by incubation with primary antibodies in TBST buffer overnight at 4 °C. Next day, the membrane was incubated with HRP-conjugated goat anti-rabbit secondary antibody (1:5000, Abcam, Cambridge, MA, USA) for 1 h at room temperature. Finally, the Electrochemiluminescence (ECL) substrate developer (Millipore) was added to membranes, exposed, and imaged by ChemiDoc Imaging System (Biorad, Hercules, CA, USA). Primary antibodies used for Western blotting were: WDR45B (Novus Biologicals (Cambridge, UK), 1:1000, Rabbit); LC3 (proteintech, 1:2500, Rabbit); p62 (CST, 1:1000, Rabbit); Akt (CST, 1:1000, Rabbit); p-Akt (Ser473) (CST, 1:2000, Rabbit); mTOR (CST, 1:1000, Rabbit); p-mTOR (Ser2481) (CST, 1:1000, Rabbit); and β-actin (CST, 1:5000, Rabbit).

### 2.6. CCK-8 Assay

Approximately 3 × 10^3^ cells suspended in 100 μL DMEM medium were seeded into a 96-well plate. A 10 μL CCK8 solution (Biosharp) was added to each well at 0 h, 24 h, 48 h, and 72 h after the cells adhered to the wall. Continued culture for 2 h was performed at 37 °C. Then, cell viability was measured using the CCK8 and microplate reader at a wavelength of 450 nm. The cell viability curve was plotted according to the absorbance values.

### 2.7. Wound-Healing Assay

Before the experiment, the backs of the 6-well plates were labeled with horizontal lines using marker pens. Each well was seeded with about 5 × 10^5^ cells and cultured in an incubator with 5% CO_2_ at 37 °C. The monolayer was scratched with a 200 μL micropipette tip, washed with PBS to remove floating cells, and cultured with fresh serum-free medium. Images were captured at 0, 24, and 48 h after the initial scratch, and the widths of scratches were measured by ImageJ software.

### 2.8. Transwell Cell Migration and Invasion Assay

Cell suspension (200 μL; 2 × 10^4^ cells/well) was added into the upper chamber of a 24-well Transwell plate with 8μm pore size (Corning, NY, USA) in migration assay or into chambers coated with Matrigel for the invasion assay. Then, 500 μL of DMEM with 20% FBS was added into the lower chamber. After 24 h or 48 h of incubation in migration assay or invasion assay, respectively, the residual cells attached to the upper surface of the membrane were gently wiped away with a cotton swab. Cells on the lower surface of the membrane were fixed with 4% paraformaldehyde for 30 min followed by staining using 0.1% (*w*/*v*) crystal violet (Sigma) for 20 min. After rinsing with PBS the stained samples were observed with an optical microscope, and the number of stained cells penetrating the pores was measured by ImageJ software (NIH, Bethesda, MD, USA).

### 2.9. Statistics Analysis

All statistical analysis was performed by using R version 4.2.0 software, IBM SPSS Statistics 23 software, and GraphPad Prism 8 software. The correlation between the staining intensity of HCC tissue microarray and matched pathological characteristics was analyzed by Chi-squared test. Comparisons between groups were performed using a two-tailed Student *t*-test. All experiments in this work were repeated for three times at least. The *p* < 0.05 was considered as statistically significant (* *p* < 0.05, ** *p* < 0.01, *** *p* < 0.001, # *p* < 0.0001).

## 3. Results

The workflow of our study is shown in Figure 1.

### 3.1. Identifying WDR45B as Prognostic Autophagy-Related Gene

It has been reported that 81 differentially expressed autophagy-related genes were screened out based on the RNA-seq transcription data of 424 HCC patients from TCGA cohort [20,21]. Among these, Univariate Cox regression analysis was performed with *p* < 0.01 as the cut-off value. It identified 29 autophagy-related genes highly correlated with survival, while 17 genes were identified in ICGC (Figure 2A), Complementary datasets from ICGC cohorts were also involved in the analysis. It turned out that there were 17 genes related to prognosis. All those genes were considered risk factors (Hazard ratio > 1) (Figure 2B). Taking the Insertion of prognostic autophagy-related genes from TCGA and ICGC cohorts, 14 key prognostic autophagy-related genes were selected (SPNS1, WDR45B, NPC1, NRAS, HSP90AB1, RHEB, GAPDH, HDAC1, BAK1, ATIC, FKBP1A, RRAGD, CDKN2A, BIRC5) (Figure 2C). The protein–protein interaction network was shown in Figure 2E. According to the retrieval from PubMed directed at these 14 genes, we chose WD repeat domain 45B (WDR45B), which has hardly been studied in HCC, as our target gene.

### 3.2. Genomic Characteristics of WDR45B in HCC

The expression levels of WDR45B in tumor tissues and adjacent normal tissues from TCGA were compared in total and in pairs (Figure 3A). It suggested that WDR45B was overexpressed in tumor tissues. The somatic mutation of WDR45B in HCC was VUS (variant of uncertain significance) at S216 site (Figure 3B). The gain of WDR45B was the main copy number variations in WDR45B in HCC (Figure 3C). GRB2 (growth factor receptor bound protein 2) (R = 0.61, *p* < 0.05) and TRIM37 (tripartite motif containing 37) (R = 0.58, *p* < 0.05) were strongly associated with WDR45B in HCC (Figure 3D). Moreover, methylation level of WDR45B in HCC was also analyzed and presented in a violin plot (Figure 3E). It demonstrated that the methylation level of WDR45B in HCC was high, almost all methylation sites were completely methylated, and only a few were unmethylated. The β ≥ 0.6 was considered completely methylated, β ≤ 0.2 was considered completely unmethylated, and 0.2 < β < 0.6 was considered partially methylated. Based on the survival data of these HCC patients, Kaplan–Meier survival analysis of WDR45B methylation level was conducted (Figure 3F), which revealed the correlation between methylation level of 9 methylation sites and related survival situations. High methylation levels of 5 methylation sites (cg05554594, cg06938133, cg1165294, cg23713156, cg03247412) suggested shorter survival time, while low methylation levels of 4 methylation sites (cg01155404, cg12167135, cg14060471, cg25363258) could predict poor prognosis.

### 3.3. Clinicopathological Characteristics and Enriched KEGG Pathways of Upregulated WDR45B in HCC

Based on the detailed clinical traits of HCC patients in TCGA cohorts involving age, sex, cirrhosis scores, grade, TNM stage, vascular invasion, and adjacent inflammation from UCSC Xena (Appendix A), the correlation between expression level of WDR45B and clinical traits was analyzed. It turned out that advanced grade and T stage was associated with high expression level of WDR45B (Figure 4A). In addition, Kaplan–Meier survival curve revealed that high expression level of WDR45B predicted poor prognosis (Figure 4B). The top 50 KEGG pathways that upregulated WDR45B were associated with were obtained through GSEA with criteria of NES > 1 and *p* < 0.05. Among these KEGG pathways, tumor-related pathways included ubiquitin-mediated proteolysis, insulin signaling pathway, cell cycle, regulation of autophagy, mTOR signaling pathway, ERBB signaling pathway, WNT signaling pathway, and Notch signaling pathway (Figure 4C).

### 3.4. WDR45B Upregulated in FFPE Tissues of HCC Patients

IHC staining of WDR45B was performed using 56 FFPE tissues of primary HCC patients and 3 normal liver FFPE tissues from patients with hepatic blunt trauma. Representative histological images of 4×, 10×, and 20× magnifications were exhibited to visualize the WDR45B expression in tumor and normal liver tissues (Figure 5A). More images are showed in the Appendix A. Moderate to strong cytoplasmic staining was visible in tumor cells, and weak positive staining can be observed in normal hepatocytes. Next, IHC staining Integrated Optical Density (IOD) values were measured and analyzed by ImagePro Plus software as well (Figure 5B). According to the median value of IHC staining scores, we divided the patients into high and low WDR45B expression groups. The correlation between WDR45B expression level and clinical pathological characteristics including age, sex, T stage, macroscope vein invasion, microvascular invasion, cirrhosis, tumor involvement, and Alpha-fetoprotein (AFP) IHC staining was analyzed by chi-square test (Table 1). It concluded that there were significant differences between grade and WDR45B IHC staining intensity in HCC patients (χ^2^ = 6.842, *p* < 0.05).

### 3.5. WDR45B Upregulated in HCC Cell Lines

Quantitative real-time PCR and Western blotting were executed to detect the WDR45B expression in HCC cell lines (HepG2 and Huh7 cell) and normal hepatocyte cells (LO2 cell). The results indicated that WDR45B expression was upregulated in HepG2 and Huh7 cells by contrast to that in LO2 cells, and more highly expressed in HepG2 cells compared with Huh7 cells (Figure 5C,D).

### 3.6. Knockdown of WDR45B Expression Inhibited Autophagy

For the further study of WDR45B in HCC, we generated HepG2 cells with stable knockdown of WDR45B using lentiviral transduction of shRNA (Figure 6A). We measured protein expression of autophagy marker LC3 and p62/SQSTM1 in WDR45B-knockdown HepG2 cells (Figure 6B). LC3-II distinctly downregulated and the ratio of LC3-II/LC3-I decreased, while p62/SQSTM1 upregulated. All these data indicated that the process of LC3-I transforming into LC3-II was blocked and p62 was accumulated, which demonstrated that autophagy was inhibited.

### 3.7. Knockdown of WDR45B Expression Suppressed Akt/mTOR Signaling Pathway

In order to investigate the potential pathway mechanism of WDR45B in HCC, we analyzed the expression of Akt, mTOR, p-Akt (Ser473), mTOR, and p-mTOR (Ser2481) after the knockdown of WDR45B (Figure 6B). There was no significant difference in the protein translation level of Akt and mTOR, while the phosphorylation level of these two proteins increased. For validating WDR45B leading to the suppression of autophagy and Akt/mTOR signaling pathway, we used 20 nM autophagy inducer rapamycin that was also a kind of mTOR inhibitor. Detecting the expression level of protein above, we found that rapamycin could reverse the expression changes partially resulting from the knockdown of WDR45B (Figure 6C). Therefore, we speculated that WDR45B might promote autophagy by activating Akt/mTOR signaling pathway.

### 3.8. Knockdown of WDR45B Expression Suppressed Proliferation and Migration in HCC

To explore the effects of WDR45B in HCC tumorigenesis, some in vitro analyses were performed. Cell viability in the WDR45B-knockdown group was significantly lower than the control group at 48 h and 72 h (Figure 7A). As for migration ability, the wound width in the WDR45B-knockdown group was markedly higher than control group at 24 h and 48 h (Figure 7B). There were some cells that obviously migrated to the wound region at 48 h in control group while there was no similar situation in the WDR45B-knockdown group (Figure 7C). In Transwell cell migration and invasion assay, less-stained cells were found in the WDR45B-knockdown group than in after 24 h or 48 h of incubation, indicating the slower migration and weaker invasion (Figure 7D). The number of cells that migrated from the WDR45B-knockdown group was significantly smaller than that in the control group through statistical analysis. Meanwhile, fewer staining cells were observed in the WDR45B-knockdown group comparing to the control group in invasion assay. All results above suggested that the migration ability of HCC cell line declined after the knockdown of WDR45B expression.

## 4. Discussion

WD repeat domain 45B (WDR45B) is a member of the WD-repeat protein interacting with the phosphoinositides (WIPIs) family, alias WIPI3 or WDR45L. WDR45B contains seven WD-repeat sequences, which are believed to fold into β-propeller structures to mediate protein–protein interactions and to constitute a conserved motif for interaction with phospholipids. The WIPIs protein family consists of four members: WIPI1, WIPI2, WDR45B/WIPI3, and WDR45/WIPI4. All of them are considered to be crucial in the autophagy process, serving as the effectors of autophagy-specific phosphatidylinositol 3-phosphate (Ptdlns3P), playing an important role in the recognition and decoding of Ptdlns3P signals in newborn autophagosomes [22]. Moreover, the WIPIs can also function as the scaffold for recruiting vital proteins or complexes, contributing to the nucleation and amplification of autophagosome membranes. The WIPIs are localized to the endoplasmic reticulum associated intima and newborn autophagosome membrane [23]. Mutations in WIPIs could seriously impact autophagy, and are closely correlated with cancers, neuronal degeneration, and intellectual disability. WIPI1 is thought to be associated with osteosarcoma, nasopharyngeal carcinoma, melanoma, and other diseases [24,25,26]. WIPI2 participates in the genesis and extension of isolated membrane, being considered to be a key downstream substrate for mTOR regulating autophagy [27,28]. Mutations in WDR45B and WDR45 lead to β-propeller protein-associated neurodegeneration and intellectual disability, respectively [29].

Autophagy starts when ATG12 conjugates with ATG5 with the assistance of ATG7 and ATG10, which is then stabilized by ATG16L and further forms an ATG12-ATG5-ATG16L complex about 800 kDa located on the outer surface of the autophagosome membrane, thus promoting the formation of LC3 conjugation system. In response to LKB1 mediated AMPK stimulation, WDR45-ATG2 can be released from the WDR45-ATG2/AMPK-ULK1 complex and transferred to the nascent autophagosome to control the size of autophagosome. The complex of WDR45B and FIP200 is also involved in this process. WDR45B receives the regulations from AMPK, which binds to the activated tuberous sclerosis complex (TSC) to control the activity of mTOR in lysosomal compartment and combines with RB1CC1/FIP200 on the nascent autophagosome to promote the nucleation and extension of autophagosome. WDR45B deficiency has been proven to obstruct the generation of newborn autophagosomes in the downstream of LC3 [29]. Related human genetic studies have revealed that the autophagy defects caused by WDR45B deletion can lead to the accumulation of SQSTM1, ubiquitin aggregates, and autophagosomes in the damaged neurons, mainly in the cerebral cortex, corpus callosum, inner sac, thalamus, and other extensive regions, destroying the neuronal and axonal homeostasis. WDR45B-knockout mice showed dyskinesia, learning, and memory deficits [30]. Recently, WDR45B was found to improve airway remodeling and reduce collagen deposition and airway hyperreactivity in mouse asthma models, by means of combining with glucocorticoid induced 1 (GLCCI1) to inhibit the autophagy activation effect of GLCCI1 [31]. At present, the majority of studies about WDR45B almost all focused on neuronal degeneration diseases, while very few on the aspect of cancer. Therefore, this study was dedicated to the potential cancer-promoting mechanism of WDR45B in HCC, hoping to bring new strategies for the diagnosis and treatment of HCC.

In our study, we found that the genomics of WDR45B was mainly characterized by the gain of copy number and abundant DNA methylation sites. The methylation level also had a certain predictive effect on the prognosis of HCC patients. In research on thioacetamide mouse liver cancer model, CpG island hypermethylation of WDR45B and Yin Yang 1 (YY1) was observed in the pretumor liver disease foci of mice, which was involved in the regulation of cell cycle and apoptosis, promoting tumor formation [32]. Furthermore, genomic analysis revealed that WDR45B was significantly co-expressed with GRB2 and TRIM37. GRB2 is considered to be a susceptibility gene for Alzheimer’s disease, and overexpression of GRB2 in prostate cancer suggests poor prognosis [33]. The SH2 domain of GRB2 binds to the insulin receptor substrates and activates tyrosine kinase, which is critical for the activation of RAS/MAPK signaling pathways [34]. In HCC, the deubiquitination enzyme PSMD14 can promote the growth and metastasis of HCC by stabilizing GRB2 [35]. As for TRIM37, it is a member of the tripartite motif containing (TRIM) family with E3 ubiquitin ligase activity. Ubiquitin mediated by TRIM37 can stabilize PEX5 and enhance the activity of peroxisome. The amplification of the TRIM37 genome contributes to the tumor progression in colorectal cancer, HCC, lung cancer, neuroblastoma, breast cancer, pancreatic cancer, and osteosarcoma [36,37,38]. It was confirmed that the overexpression of TRIM37 in HCC could promote the invasion and strengthen sorafenib resistance by activating the AKT signaling pathway [39].

According to the outcomes of GSEA the upregulation of WDR45B significantly enriched the mTOR signaling pathway, which was validated by experiments in our study. Autophagy marker LC3 is a light chain protein, mainly involved in the formation of autophagosomes. LC3 precursor molecules are cleaved by ATG4B at the carboxyl-terminal to form LC3-I, which is then conjugated with lipoylethanolamine to generate lipidized LC3-II, attaching to the autophagosome membrane as the structural skeleton [40]. The LC3-II/LC3-I ratio is extensively applied to monitor autophagy levels. P62/SQSTM1 as a pivotal protein taking part in autophagy-lysosome and ubiquitin protease systems can be degraded by proteolytic enzymes during lysosomal hydrolysis with the interaction with LC3 [41]. Autophagy defects are a common upregulation mechanism of P62/SQSTM1 in human tumors [42]. In our study, we confirmed in the HCC cell line that WDR45B knockdown could significantly inhibit autophagy by reducing LC3-II, increasing the accumulation of P62/SQSTM1, and raising the phosphorylation levels of Akt and mTOR. It indicated that WDR45B might promote autophagy by inhibiting the Akt/mTOR signaling pathway in HCC. At present, a large volume of research has shown that autophagy can maintain the survival of tumor cells by providing nutrition and energy under metabolic and oxidative stress originating from anti-cancer therapy in established metastatic tumors, which is a crucial mechanism of drug resistance to anti-cancer therapy in patients with advanced HCC. In HCC, the emergence of intrinsic acquired drug resistance to sorafenib is still a huge challenge for the prognosis of patients with advanced HCC, where only about 30% of patients respond to sorafenib treatment [43]. Autophagy activation caused by the inhibition of Akt/mTOR signaling pathway is an important mechanism of sorafenib drug resistance [44]. Cell surface molecules such as CD24, BEZ235, and SIRT1 have been found to suppress mTOR and promote autophagy in HCC, leading to resistance to sorafenib [45,46,47]. In addition to sorafenib, increased autophagy flux has also been observed in other anti-cancer drugs for HCC. Inhibition of autophagy has been shown to enhance the sensitivity of sorafenib and other anticancer drugs to a certain extent, and bring substantial survival benefits to patients. Currently, autophagy inhibitors chloroquine and hydroxychloroquine have been used clinically, as they can deacid lysosomes and block the fusion between autophagosomes and lysosomes preventing cargo degradation. Chloroquine can also sensitize tumor cells to chemotherapy drugs independent of autophagy approaches. Other autophagy modulators, such as VPS34, ULK1, and ATG4B inhibitors have been confirmed to exert tumor suppressor effect in clinical mouse models [48].

## 5. Conclusions

In conclusion, our study demonstrates that WDR45B is upregulated in HCC, suggesting less clear differentiation and poor prognosis. In vitro experiments showed that the knockdown of WDR45B can suppress autophagy by upregulating the Akt/mTOR signaling pathway and reduce tumor proliferation and migration. Our study provides a new insight to the mechanism of Akt/mTOR signaling pathway, suggesting that selective Akt/mTOR signaling pathway inhibitors may help to overcome drug resistance and improve the efficacy of anti-cancer treatment.

## Figures and Tables

**Figure 1 diagnostics-13-00906-f001:**
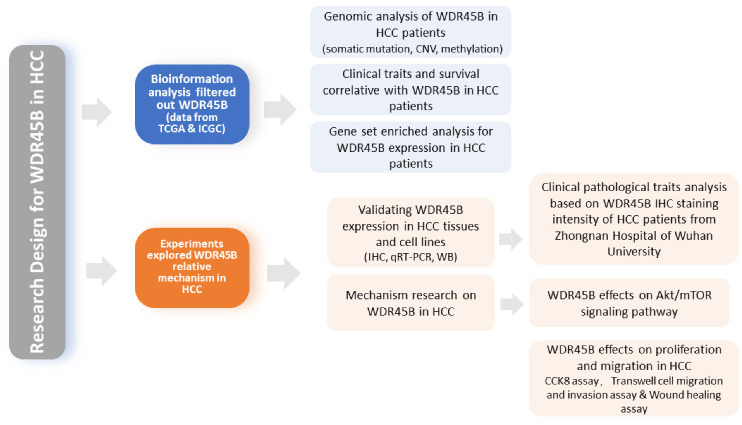
Workflow of research design for WDR45B in HCC.

**Figure 2 diagnostics-13-00906-f002:**
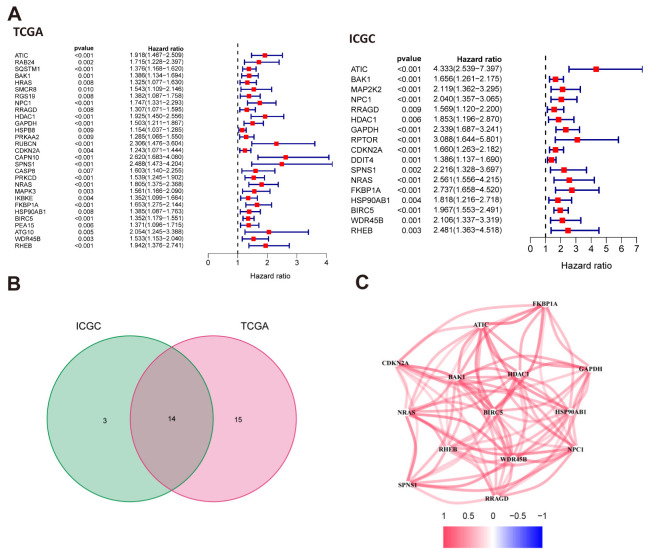
Searching for autophagy-related genes associated with HCC prognosis. (**A**) Univariate Cox regression analysis of autophagy-related genes associated with prognosis in HCC patients from TCGA and ICGC databases; (**B**) 14 autophagy-related prognostic genes were obtained by intersection; (**C**) The correlation between 14 autophagy-related prognostic genes. HR, hazard ratio; *p* < 0.05 with significantly difference.

**Figure 3 diagnostics-13-00906-f003:**
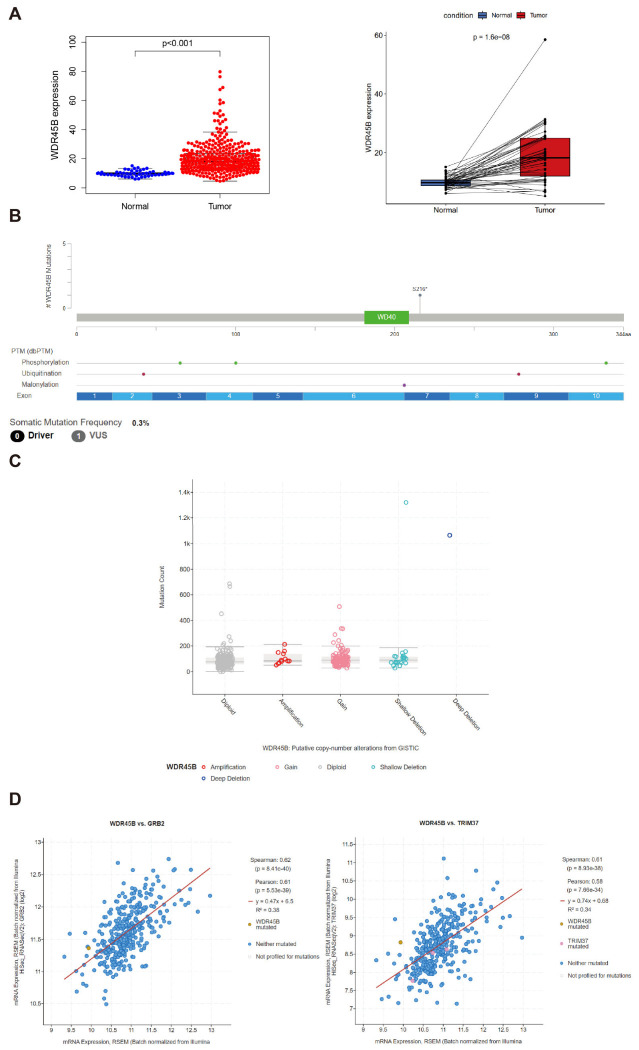
Genomic characteristics of WDR45B in HCC patients. (**A**) Relative expression level of WDR45B in HCC patients based on TCGA database; (**B**) The somatic mutation of WDR45B in HCC patients from TCGA cohorts; (**C**) The copy number variation in WDR45B in HCC patients based on TCGA database; (**D**) GRB2 (R = 0.61, *p* < 0.05) and TIM37 (R = 0.58, *p* < 0.05) co-expression with WDR45B in HCC patients from TCGA cohorts; (**E**) DNA methylation level of WDR45B in HCC patients from TCGA cohorts; (**F**) Kaplan–Meier survival curve in relation to WDR45B methylation status in HCC patients from TCGA cohorts. VUS, variant of uncertain significance; R, Pearson correlation coefficient.

**Figure 4 diagnostics-13-00906-f004:**
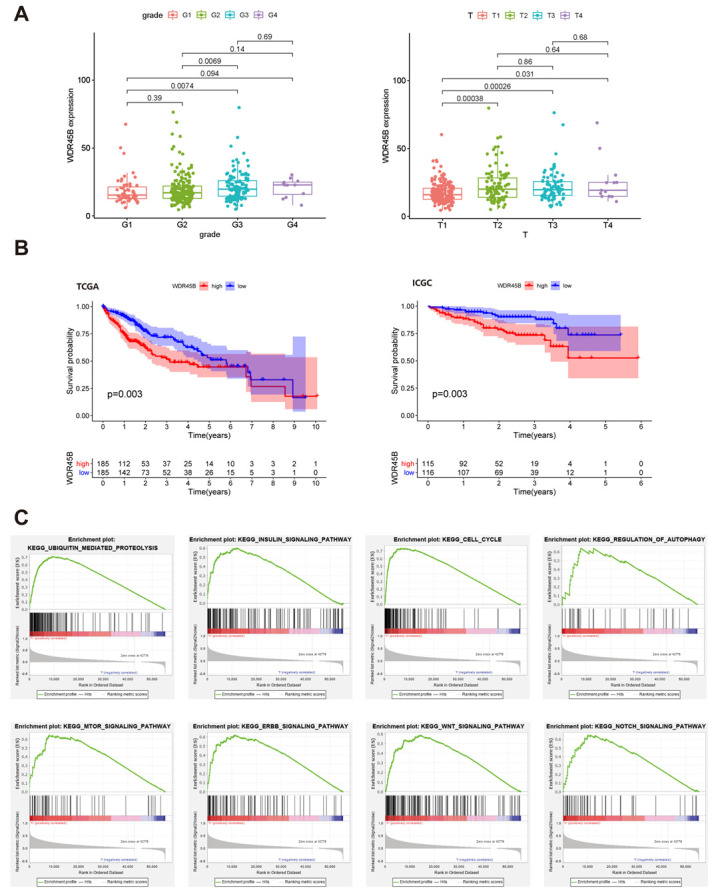
Clinicopathological characteristics and enriched KEGG pathway correlative with WDR45B in HCC patients. (**A**) The expression level of WDR45B associated with grade and T stage in HCC patients (*p* < 0.05); (**B**) Kaplan–Meier survival curve of WDR45B expression in HCC patients from TCGA and ICGC cohorts; (**C**) KEGG pathway that overexpressed WDR45B enriched in between HCC patients (NES > 1, *p* < 0.05): ubiquitin-mediated proteolysis, insulin signaling pathway, cell cycle, regulation of autophagy, mTOR signaling pathway, ERBB signaling pathway, WNT signaling pathway, and notch signaling pathway.

**Figure 5 diagnostics-13-00906-f005:**
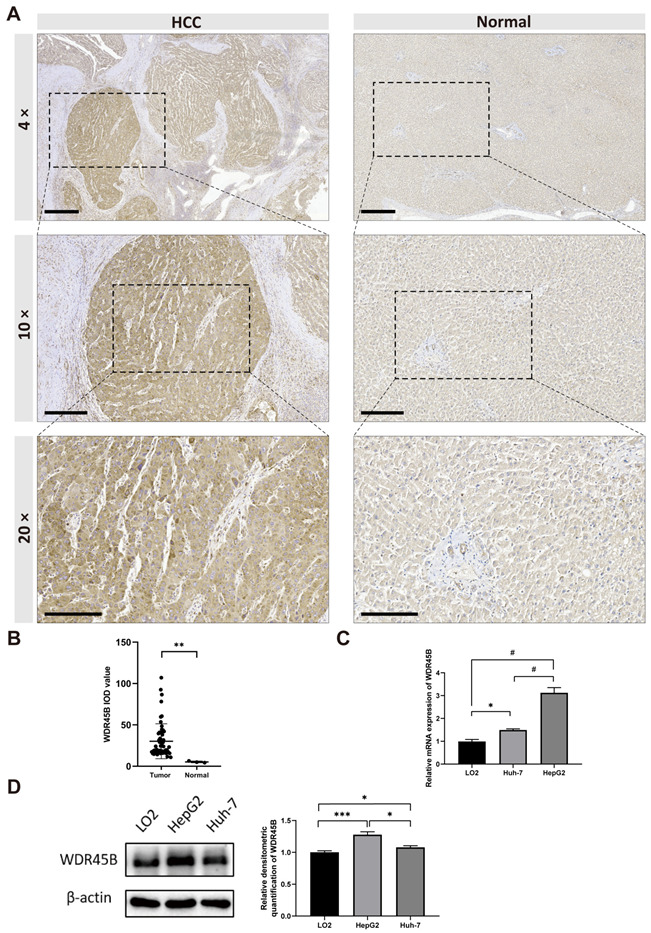
The expression of WDR45B in HCC. (**A**,**B**) Immunohistochemistry analysis of WDR45B overexpressed in HCC tissues (Grade: G3, T stage: T3) compared with normal liver tissue (magnification ×4, ×10, ×20); (**C**) qRT-PCR analysis of WDR45B relative mRNA expression in LO2, Huh-7, and HepG2 cell lines; (**D**) Western blot analysis of WDR45B expression levels in LO2, HepG2, and Huh-7 cell lines. Scale bar: 600 μm (magnification ×4), 300 μm (magnification ×10), 200 μm (magnification ×20); * *p* < 0.05, ** *p* < 0.01, *** *p* < 0.001, # *p* < 0.0001.

**Figure 6 diagnostics-13-00906-f006:**
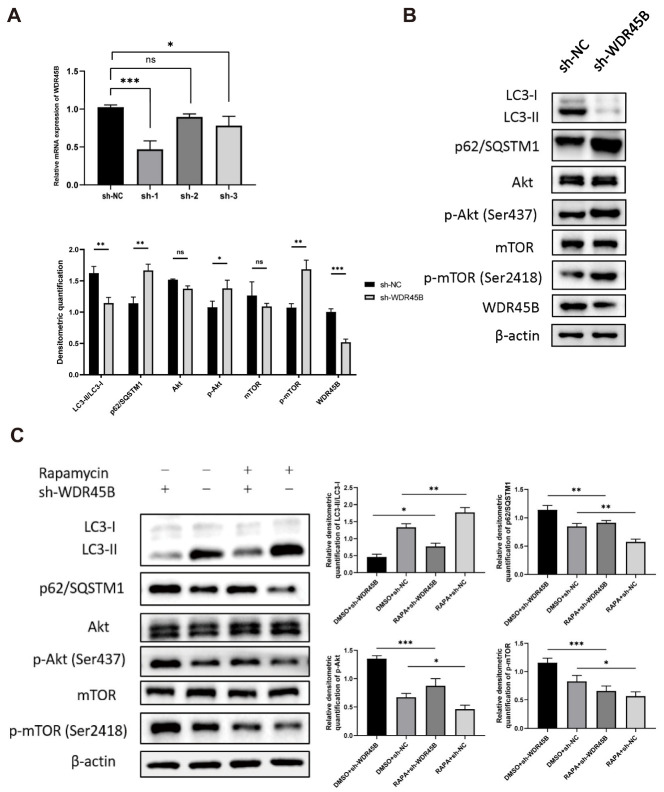
Knockdown WDR45B effected autophagy and Akt/mTOR signaling pathway in HCC. (**A**) shRNA knockdown efficiency validated by qRT-PCR analysis; (**B**) Expression level of autophagy markers (LC3, p62/SQSTM1) and Akt/mTOR signaling pathway relative proteins (Akt, mTOR, p-Akt, p-mTOR) measured by Western blot; (**C**) Expression level of autophagy markers and Akt/mTOR signaling pathway relative proteins after rapamycin treatment (20 nM, 24 h) measured by Western blot, 0.1% DMSO as negative control. ns, no statistical difference; * *p* < 0.05, ** *p* < 0.01, *** *p* < 0.001.

**Figure 7 diagnostics-13-00906-f007:**
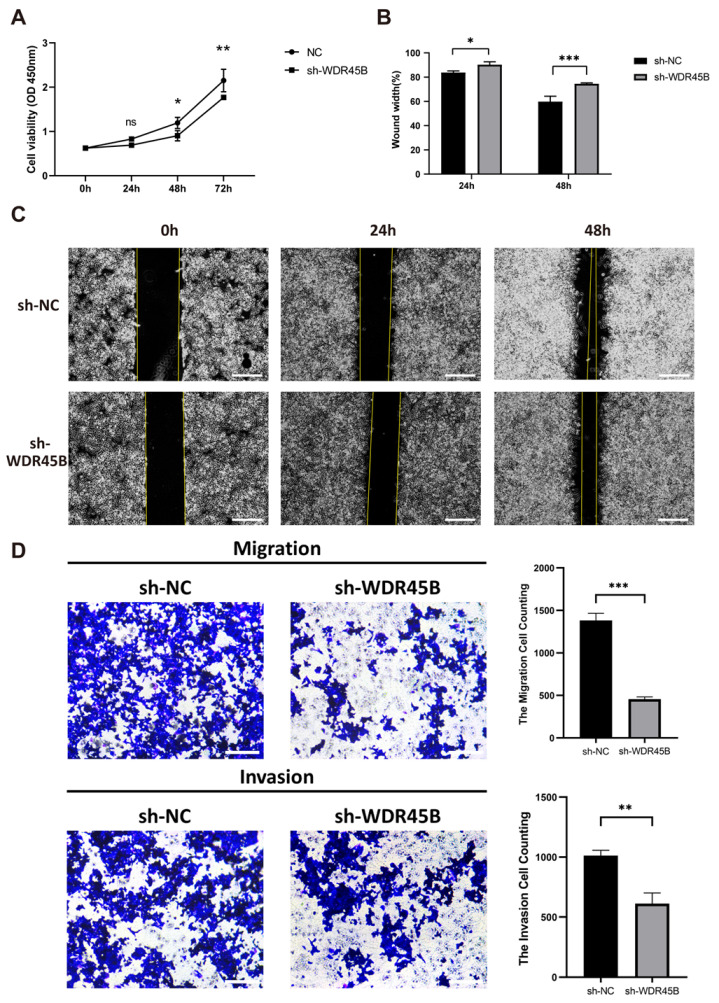
Knockdown WDR45B declined proliferation and migration of HCC. (**A**) CCK8 assay indicated that WDR45B knockdown reduced viability; (**B**,**C**) Wound healing assay demonstrated that WDR45B knockdown inhibited cell migration; (**D**) Transwell cell migration and invasion assay showed that WDR45B knockdown inhibited cell migration ability. Scale bar: 500 μm (**C**), 200 μm (**D**); ns, no statistical difference; * *p* < 0.05, ** *p* < 0.01, *** *p* < 0.001.

**Table 1 diagnostics-13-00906-t001:** Relationship between clinical pathological traits and WDR45B expression in HCC patients from Zhongnan Hospital of Wuhan University.

Variables		Total HCC Patients	WDR45B Expression in HCC Patients		χ^2^	*p*
			High	Low		
Age, n (%)	>45	42 (75.0%)	24 (85.7%)	18 (64.3%)	3.429	0.121
	≤45	14 (25.0%)	4 (14.3%)	10 (35.7%)		
Sex, n (%)	male	55 (98.2%)	27 (96.4%)	28 (100%)	1.018	1.000
	female	1 (2.8%)	1 (3.6%)	0 (0%)		
Grade, n (%)	G1-G2	39 (69.6%)	15 (53.6%)	24 (85.7%)	6.842	0.019 *
	G3	17 (30.4%)	13 (46.4%)	4 (14.3%)		
T stage, n (%)	T1-T2	33 (58.9%)	13 (46.4%)	20 (71.4%)	3.615	0.102
	T3-T4	23 (41.1%)	15 (53.6%)	8 (28.6%)		
macroscope vein invasion, n (%)	visible	6 (10.7%)	4 (14.3%)	2 (7.1%)	0.747	0.669
	invisible	50 (89.3%)	24 (85.7%)	26 (92.9%)		
microvascular invasion, n (%)	visible	36 (64.3%)	20 (71.4%)	16 (57.1%)	1.244	0.403
	invisible	20 (35.7%)	8 (28.6%)	12 (42.9%)		
cirrhosis, n (%)	yes	22 (39.3%)	12 (42.9%)	10 (35.7%)	0.299	0.785
	no	34 (60.7%)	16 (57.1%)	18 (64.3%)		
tumor involvement, n (%)	restricted to liver	47 (88.7%)	23 (85.2%)	24 (92.3%)	0.669	0.669
	invade portal vein, gallbladder or visceral peritoneum	6 (11.3%)	4 (14.8%)	2 (7.7%)		
AFP (IHC), n (%)	positive (+)	16 (61.5%)	9 (75.0%)	7 (50.0%)	1.706	0.248
	negative (−)	10 (38.5%)	3 (25.0%)	7 (50.0%)		

* *p* < 0.05.

## Data Availability

The datasets generated during the current study are available in TCGA (The Cancer Genome Atlas, https://portal.gdc.cancer.gov/, accessed on 30 April 2021), UCSC Xena (http://xena.ucsc.edu/, accessed on 30 April 2021), ICGC (International Cancer Genome Consortium, https://dcc.icgc.org/, accessed on 30 April 2021), HADb (Human Autophagy Database, http://www.autophagy.lu/, accessed on 4 May 2021), Pubmed (https://pubmed.ncbi.nlm.nih.gov/, accessed on 15 June 2021), cBioPortal for Cancer Genomics (http://www.cbioportal.org/, accessed on 2 May 2022), MethSurv (https://biit.cs.ut.ee/methsurv/, accessed on 21 August 2021).

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
