# Peer review of "Autophagy-Related Gene WD Repeat Domain 45B Promotes Tumor Proliferation and Migration of Hepatocellular Carcinoma through the Akt/mTOR Signaling Pathway"

_diagnostics, 2023, doi:10.3390/diagnostics13050906_

Round 1
Reviewer 1 Report
Using public databases, the authors identified WDR45B as key gene among those involved in the autophagic process in HCC. Then, using in vitro silencing experiments they identified which pathways are regulated by this protein, suggesting WDR45B as a new biomarker for HCC.
The methods and study design are appropriate for answering the research question. The manuscript is well organized, and the language is clear and understandable.
I have some comments and suggestions for the manuscript, listed below:
· Figure 1 A, B is already published in the manuscript “Construction of a prognosis-predicting model based on autophagy- related genes for hepatocellular carcinoma (HCC) patients” (Yayun Zhu, Ru Wang, Wanbin Chen, Qiuyu Chen, Jian Zhou. Aging 2020) and in the manuascript “Identification and Validation of a Prognostic Model Based on Three Autophagy-Related Genes in Hepatocellular Carcinoma” (Fanbo Qin, Junyong Zhang, Jianping Gong, and Wenfeng Zhang. BioMed Research International 2021). Please add these references in the text and edit the manuscript by deleting this part
· Improve the resolution of the figures and enlarge the character especially in Figures 2,3,4. I suggest dividing Figure 3 into two separate figures. Figure 3 panel F is completely unreadable
· figure legends need to be better explained. Moreover, many panels consist of several graphs. Please explain each one. For example, in Figure 3 panel A: what is the difference between the graphs on the left and right? are the same patients graphed?
· Page 2 lines 63-64: are you sure that macro-autophagy is called macrophage?
· Page 3 lines 121-122: which program did you use to calculate the protein-protein interaction?
· Adding access dates to the various databases
· Page 5 line 207: are you sure you used the 200 ul tip? what is the size of the gap you got?
· How were GRB2 and TRIM37 identified?
· Page 9 line 290: are the patients 58 or 56?
· Table 1: what was the criterion for dividing the patients into low and high WDR45B expression?
· Explain all abbreviations, such as AFP. Is it Alpha fetoprotein?
· Figure 5 panel A: from which patient are the tissues derived? specify stage and grade
· Figure 5 panel B: is the average of all analyzed samples represented? is the difference statistically significant?
· Figure 5 panel D: is the difference between LO2 and Huh7 statistically significant? Please check the analysis again
· Page 12 line 322: LC3II is downregulated. Moreover, regarding LC3 expression, it is true that LC3II is downregulated, but from the western blot the impression is that there is a total decrease in the levels of this protein. Did the authors check the mRNA levels? the densitometry regarding LC3II/LC3I does not seem realistic to me compared to what is seen in the blot
· I would also like to see the western blot of the WDR45B silencing added to the Figure 6 panel B
· Page 13 line 346: CCK8 estimates cell viability and not proliferation. If the authors want to characterise proliferative capacity, they should perform a test such as BrdU incorporation
· Page 13 line 348: “higher than control” From the images, it is not so evident
· Figure 7 panel A: add the values at t 0. Furthermore, do the values represented correspond to the average OD?
· Figure 7 panel B: are the values on the y-axis %? Check
· Ref 27 and 28 are the same
· Regarding the migratory capacity, to give more emphasis to the data I suggest to perform also invasion assay. And, given that tumor invasion and metastasis depend on EMT, authors could analyze the levels of EMT markers (E-cadherin, vimentin, N-cadherin…) to assess the effect of WDR45B knockdown on EMT.
Author Response
Response to Reviewer 1 Comments
Using public databases, the authors identified WDR45B as key gene among those involved in the autophagic process in HCC. Then, using in vitro silencing experiments they identified which pathways are regulated by this protein, suggesting WDR45B as a new biomarker for HCC.
The methods and study design are appropriate for answering the research question. The manuscript is well organized, and the language is clear and understandable.
I have some comments and suggestions for the manuscript, listed below:
Point 1: Figure 1 A, B is already published in the manuscript “Construction of a prognosis-predicting model based on autophagy- related genes for hepatocellular carcinoma (HCC) patients” (Yayun Zhu, Ru Wang, Wanbin Chen, Qiuyu Chen, Jian Zhou. Aging 2020) and in the manuascript “Identification and Validation of a Prognostic Model Based on Three Autophagy-Related Genes in Hepatocellular Carcinoma” (Fanbo Qin, Junyong Zhang, Jianping Gong, and Wenfeng Zhang. BioMed Research International 2021). Please add these references in the text and edit the manuscript by deleting this part
Response 1: Much obliged to your patience and carefulness. Reletive content has been revised and references have been added.
Point 2: Improve the resolution of the figures and enlarge the character especially in Figures 2,3,4. I suggest dividing Figure 3 into two separate figures. Figure 3 panel F is completely unreadable
Response 2: The resolution of the Figure 2,3,4 has been improved and Figure 3 has been seperated in two pictures.
Point 3: Figure legends need to be better explained. Moreover, many panels consist of several graphs. Please explain each one. For example, in Figure 3 panel A: what is the difference between the graphs on the left and right? are the same patients graphed?
Response 3: Every graph has been explained in Results part or figure legends now.
Point 4: Page 2 lines 63-64: are you sure that macro-autophagy is called macrophage?
Response 4: We thank the reviewer’s carefulness. Vocabulary errors have been corrected in the revised manuscript.
Point 5: Page 3 lines 121-122: which program did you use to calculate the protein-protein interaction?
Response 5: We use “STRINGdb” R package to obtain data and “igraph” R package to visualize the protein-protein interation network. Related content has been added in revised manuscript.
Point 6: Adding access dates to the various databases
Response 6: Access dates have been added in Methods part.
Point 7: Page 5 line 207: are you sure you used the 200 ul tip? what is the size of the gap you got?
Response 7: We used the 200 ml tip and the size of the gap was around 0.73 mm. ImagJ software was used to measure the width.
Point 8: How were GRB2 and TRIM37 identified?
Response 8: GRB2 and TRIM37 were the top 2 genes most strongly associated with WRD45B based on the TCGA-HCC cohorts from cBioPortal for Cancer Genomics (http://www.cbioportal.org/). This missing part has been added to the revised manuscript.
Point 9: Page 9 line 290: are the patients 58 or 56?
Response 9: The total number was mistaked during data process, it has now been corrected as 56.
Point 10: Table 1: what was the criterion for dividing the patients into low and high WDR45B expression?
Response 10: We divided the patients into high and low WDR45B expression groups according to the median value of IHC Integrated Optical Density (IOD).
Point 11: Explain all abbreviations, such as AFP. Is it Alpha fetoprotein?
Response 11: Abbreviations have been explained now in the revised manuscript.
Point 12: Figure 5 panel A: from which patient are the tissues derived? specify stage and grade
Response 12: Graphs were from the patient with Grade 3 and T stage III. Relative content has been added in the figure legends.
Point 13: Figure 5 panel B: is the average of all analyzed samples represented? is the difference statistically significant?
Response 13: We used student t-test to analyze the samples rather than simply comparing the average value. New visualization form has been chosen to demonstrate the significant difference better.
Point 14: Figure 5 panel D: is the difference between LO2 and Huh7 statistically significant? Please check the analysis again
Response 14: The analysis is correct after double check.
Point 15: Page 12 line 322: LC3II is downregulated. Moreover, regarding LC3 expression, it is true that LC3II is downregulated, but from the western blot the impression is that there is a total decrease in the levels of this protein. Did the authors check the mRNA levels? the densitometry regarding LC3II/LC3I does not seem realistic to me compared to what is seen in the blot
Response 15: We thank the reviewer’s carefulness. LC3-II is indeed downregulated. After checking the anlysis, we found there was an input error during the data process which leads to the wrong demontration in the “LC3-II/LC3-I” column. Writing errors and graphic errors have been corrected in the revised manuscript.
Point 16: I would also like to see the western blot of the WDR45B silencing added to the Figure 6 panel B
Response 16: Relative graph has been added to the Figure 6 panel B.
Point 17: Page 13 line 346: CCK8 estimates cell viability and not proliferation. If the authors want to characterise proliferative capacity, they should perform a test such as BrdU incorporation
Response 17: CCK8 estimates cell viability which reflects the ability of proliferation in a broad sense. However, we take the reviewer’s point and change a certain amount of “proliferation” to “viability”.
Point 18: Page 13 line 348: “higher than control” From the images, it is not so evident
Response 18: ImageJ software was used to analyze the width of these wounds. The observation from naked eyes may not seem evidently as the numeric data showed due to the limited resolution and maginification of the images.
Point 19: Figure 7 panel A: add the values at t 0. Furthermore, do the values represented correspond to the average OD?
Response 19: The values at t 0 have been added. The values represented the average OD and the error bars represented the standard deviation.
Point 20: Figure 7 panel B: are the values on the y-axis %? Check
Response 20: The values on the y-axis have been corrected in the revised manuscript.
Point 21: Ref 27 and 28 are the same
Response 21: The repeated reference has been removed in the revised manuscript.
Point 22: Regarding the migratory capacity, to give more emphasis to the data I suggest to perform also invasion assay. And, given that tumor invasion and metastasis depend on EMT, authors could analyze the levels of EMT markers (E-cadherin, vimentin, N-cadherin…) to assess the effect of WDR45B knockdown on EMT.
Response 22: Thank you for your advice. Invasion assay was performed and added in the revised manuscript. However, more EMT-related experiments are not available in our laboratory and in the limited revision time. Further exploration could be considered in the future. Thanks again.

Reviewer 2 Report
Jiahao Li etc. found a potential autophagy-related biomarker WDR45B for Hepatocellular carcinoma (HCC) based on bioinformatic analysis. They validated WDR45B expression in HCC tissues and cell lines. They further explored WDR45B-related mechanisms in HCC. They found that the knockdown of WDR45B could suppress autophagy by upregulating the Akt/mTOR signaling pathway and reducing tumor proliferation and migration. Overall, this paper provided a novel target for clinical diagnoses and treatment of HCC. It provided sufficient background, methods are adequately described and results are clearly presented.
Minor revision:
- The resolution of Figures needs to improve. For example, Figure 3 is not clear. I can’t figure out the titles, x-labels, and y-labels in panels C – F.
- In Table 1, the variable ‘Grade’ is analyzed in 2 subgroups, ‘G1-G2’ and ‘G3’. However, in Figure 4 panel A, there are 4 levels of Grade, G1, G2, G3, and G4. As the conclusion is ‘advanced grade and T stage was associated with high expression level of WDR45B’, why is G4 not in the subgroup of ‘Grade’ in Table 1?
- On line 290, it said ‘58 FFPE tissues of primary HCC patients’, why in Table 1, the total number of HCC patients is only 56?
- Could you provide the exact data source for the bioinformatic analysis? For example, UCSC Xena (http://xena.ucsc.edu/) is mentioned as the public database of clinical traits of HCC patients. But what is the study id? Can you specify the link/id of the data source used in this manuscript?
- In figure 5 panel A, are these x4, x10, and x20 representative images from the same image? If so, could you circle out which part is zoomed in from x4 and x10 images?
Author Response
Jiahao Li etc. found a potential autophagy-related biomarker WDR45B for Hepatocellular carcinoma (HCC) based on bioinformatic analysis. They validated WDR45B expression in HCC tissues and cell lines. They further explored WDR45B-related mechanisms in HCC. They found that the knockdown of WDR45B could suppress autophagy by upregulating the Akt/mTOR signaling pathway and reducing tumor proliferation and migration. Overall, this paper provided a novel target for clinical diagnoses and treatment of HCC. It provided sufficient background, methods are adequately described and results are clearly presented.
Point 1: The resolution of Figures needs to improve. For example, Figure 3 is not clear. I can’t figure out the titles, x-labels, and y-labels in panels C – F.
Response 1: We thank the reviewer’s suggestions. Figure 3 have been detached into two pictures in order to improve the resolution of these pictures.
Point 2: In Table 1, the variable ‘Grade’ is analyzed in 2 subgroups, ‘G1-G2’ and ‘G3’. However, in Figure 4 panel A, there are 4 levels of Grade, G1, G2, G3, and G4. As the conclusion is ‘advanced grade and T stage was associated with high expression level of WDR45B’, why is G4 not in the subgroup of ‘Grade’ in Table 1?
Response 2: Figure 4 panel A is an automatically produced picture from databases on the website while the content of Table 1 was based on the clinical data from our pathological archives. And Grade 4 patients were not found in our research period of time.
Point 3: On line 290, it said ‘58 FFPE tissues of primary HCC patients’, why in Table 1, the total number of HCC patients is only 56?
Response 3: Thanks for your carefulness. The total number was mistaked during data process, it has now been corrected as 56.
Point 4: Could you provide the exact data source for the bioinformatic analysis? For example, UCSC Xena (http://xena.ucsc.edu/) is mentioned as the public database of clinical traits of HCC patients. But what is the study id? Can you specify the link/id of the data source used in this manuscript?
Response 4: HCC patients data source is https://gdc-hub.s3.us-east-1.amazonaws.com/download/TCGA-LIHC.GDC_phenotype.tsv.gz, which has been added as Supplementary Table 1 in the revised manuscript.
Point 5: In Figure 5 panel A, are these x4, x10, and x20 representative images from the same image? If so, could you circle out which part is zoomed in from x4 and x10 images?
Response 5: The zoomed parts have been circled in Figure 5 panel A in the revised manuscript.

Reviewer 3 Report
Very Well written paper.
A minor suggestion from my side is to add further pictures from the IHC experiments to better display the WDR45B positivity
Author Response
Point 1: Very well written paper. A minor suggestion from my side is to add further pictures from the IHC experiments to better display the WDR45B positivity.
Response 1: Thank you for your appreciation to our manuscript. More pictures of IHC assay have been submitted as supplementary figures in the revised manuscript.

Round 2
Reviewer 1 Report
The authors replied to most of my comments. However
· some writing errors still remained, for example ”Bioinformation analysis” in line 20
· I am sorry but I still do not agree with interpretation of the LC3 data. WDR45B KD causes a decrease in LC3II, but there is no increase in the LC3I form. Moreover, the LC3II/LC3I ratio should be lower in the silenced WDR45B than in the NC
· Lines 377-380: delete “Figure 7E” and add statistics regarding invasion assay.
Author Response
The authors replied to most of my comments. However
Point 1: Some writing errors still remained, for example ”Bioinformation analysis” in line 20.
Response 1: Certain writing error has been corrected. Microsoft Word somehow did not underline this error and we thank you for your carefulness.
Point 2: I am sorry but I still do not agree with interpretation of the LC3 data. WDR45B KD causes a decrease in LC3II, but there is no increase in the LC3I form. Moreover, the LC3II/LC3I ratio should be lower in the silenced WDR45B than in the NC.
Response 2: We really appreciate your comments. The LC3II/LC3I ratio was indeed decreased after WDR45B knockdown. This mistake occurred because we switch the position of the control group and knockdown group when redecorating the visualization of these results, and the LC3 column was neglected. The correct graph has been added in the revised manuscript.
Point 3: Lines 377-380: delete “Figure 7E” and add statistics regarding invasion assay.
Response 3: Certain sentences have been modified at reviewer’s request.
